# Identification and Characterization of *Hdh-FMRF2* Gene in Pacific Abalone and Its Possible Role in Reproduction and Larva Development

**DOI:** 10.3390/biom13010109

**Published:** 2023-01-05

**Authors:** Zahid Parvez Sukhan, Yusin Cho, Shaharior Hossen, Won Kyo Lee, Kang Hee Kho

**Affiliations:** Department of Fisheries Science, Chonnam National University, Yeosu 59626, Republic of Korea

**Keywords:** FMRFamide, FMRF2, neuropeptide, reproduction, larval development, Pacific abalone, *Haliotis discus hannai*

## Abstract

FMRFamide-related peptides are neuropeptides involved in a wide range of biological processes, including reproduction and larval development. To characterize the involvement of FMRFamide in the reproduction and larval development of Pacific abalone *Haliotis discus hannai*, an FMRFamide cDNA (*Hdh-FMRF2*) was cloned from the cerebral ganglion (CG). Fluorescence in situ hybridization and qRT-PCR were performed for functional characterization. The *Hdh-FMRF2* cDNA encoded 204 deduced amino acids that contained a putative signal peptide and four FaRP domains. The major population of *Hdh-FMRF2* neuronal cell bodies was localized in the cortex of CG. *Hdh-FMRF2* mRNA expression was significantly upregulated in CG during the mature stage of gonadal development and effective accumulative temperature (EAT) exposed abalone in both sexes. In the induced spawning event, *Hdh-FMRF2* expression was significantly upregulated during spawning in males. However, no upregulation was observed in females, suggesting *Hdh-FMRF2* might inhibit gamete release in female abalone. These results revealed *Hdh-FMRF2* as a reproduction related peptide. Furthermore, mRNA expression in larval development suggested that this peptide was also involved in larval development during development of Pacific abalone. Collectively, this study provides evidence of possible involvement of an FMRFamide neuropeptide in the reproduction and larval development of Pacific abalone.

## 1. Introduction

Neuropeptides generally act as neurotransmitters, neuromodulators, and neurohormones and are widely distributed among diverse species across the animal kingdom. FMRFamide-related peptides (FaRPs) are the largest and most diverse family of neuropeptides known to be expressed in both vertebrates and invertebrates [1]. FaRPs are intimately involved in a broad array of biological processes, including metamorphosis [2,3], feeding behavior [4], cardiovascular function [5], locomotion [6], reproduction [2,7,8], homeostasis and defense [9].

FaRPs are the most comprehensively studied neuropeptides in invertebrates [10,11,12] with tetrapeptide FMRFamide (Phe-Met-Arg-Phe-NH2; FMRFa) being one of the most representative members of this peptide family. FaRPs share a high sequence similarity with FMRFamide and are evolutionarily related to FMRFamide. The first FaRP, an FMRFamide was identified in the sunray venus clam (*Macrocallista nimbosa*), a bivalve mollusk species [5]. Since then, the sequence, structure, localization and physiological effects of FaRPs have been widely studied in cephalopods, gastropods, and other species [2,7,13,14,15,16,17]. More than 40 types of FaRPs have thus far been identified in the 5 major mollusk classes [14,18]. Among gastropod mollusks, the FMRFamide gene has been studied extensively in pond snail [11,17] and sea hare [14,16]. In abalone species, two transcripts of FMRFamide have been reported in *Haliotis asinina* [2] and one transcript in Pacific abalone, *Haliotis discus hannai* (GenBank Accession No. MF066907).

Abalone are gastropod mollusks that are widely distributed in tropical and temperate waters. Among abalone species, the Pacific abalone *Haliotis discus hannai* is considered the most important commercial species in South-East Asian countries, including Korea, due to its high price and extraordinary nutritional value [19]. The edible parts of this species contain bioactive molecules with confirmed antioxidant, anticancer, anti-inflammatory, and antimicrobial properties [20,21]. However, wild abalone fishery yields have decreased dramatically in Korea and other parts of the world in the past decades [22]. Therefore, commercial open-sea cage aquaculture has rapidly expanded to meet consumer demand. Critically, these abalone cage farms require large volumes of abalone seeds, which are produced in abalone hatcheries [23]. To sustain proper reproductive function in the hatchery, the abalone broodstock must be of the best quality, with proper gonadal maturation and reproductive performance during spawning, which are of the utmost importance. Previous studies have reported that captive reared abalone attained sexual maturity earlier than their wild counterparts [24,25]. Furthermore, the metamorphosis of the hatchery produced larvae tends to be delayed compared to wild abalone larvae [26]. Therefore, it is important to study the neuroendocrine genes involved in reproduction and larval development of abalone. Several neuroendocrine genes are involved in gonadal development, maturation, and spawning of mollusk species. FMRFamide is considered a reproduction related neuropeptide in mollusk species and plays a crucial role in sexual maturation of gastropod species [2,27]. Moreover, FMRFamide is also involved in embryonic and larval development in mollusks [2,3]. To determine whether FMRFamide neuropeptide is involved in the reproduction and larval development of Pacific abalone, a second transcript of the FMRFamide gene, *Hdh-FMRF2* was isolated and characterized from the cerebral ganglion (CG) of Pacific abalone. The levels of mRNA expression of *Hdh-FMRF2* were studied in several reproductive maturation conditions and during induced spawning events of Pacific abalone. Additionally, fluorescence in situ hybridization was conducted to detect the neuronal expression of *Hdh-FMRF2* in the CG of Pacific abalone. Furthermore, *Hdh-FMRF2* mRNA expressions were monitored during the embryonic and larval development of Pacific abalone.

## 2. Materials and Methods

### 2.1. Experimental Animal and Sample Collection

Three-year-old male and female Pacific abalone (*H. discus hannai*) were collected from sea cages of the coast of Jindo-gun, Republic of Korea. The abalone had a mean body weight of 109.04 ± 0.56 g and mean shell length of 82.04 ± 0.49 cm. All collected abalone were transported to the Tou-Jeong Soosan abalone hatchery in Dolsan-eup (Yeosu-si, Jeollanam-do, Republic of Korea) and reared in conditioning tanks with running sea water.

### 2.2. Tissues Collection for Gene Cloning

After acclimatizing for 7 days, 10 abalone were brought to the Laboratory of Molecular Physiology of the Department of Fisheries Science, Chonnam National University, Republic of Korea for tissue collection. Abalone were first anesthetized with 5% MgCl_2_, then cerebral ganglia (CG) were collected for cloning of the FMRFamide gene. The CGs were then washed with 1× phosphate buffered saline (PBS), immediately flash-frozen in liquid nitrogen, and stored at −80 °C until total RNA extraction.

### 2.3. Collection of CG and Preparation of Frozen Sections for In Situ Hybridization

Abalone were first anesthetized with 5% MgCl_2_ and then CG tissues were collected for in situ hybridization experiment. The collected CGs were then washed with 1× PBS and fixed in 4% paraformaldehyde (PFA) overnight. CG tissues were then infiltrated with 30% sucrose and embedded in optimum cutting temperature (OCT) compound (FSC 222, Leica Biosystems, Wetzlar, Germany), then frozen at −20 °C overnight. The embedded tissues were then sectioned into 6 µM thick slices in transverse orientation using a cryostat microtome (Leica CM3050; Wetzlar, Germany). The sections were fixed on SuperFrost™ Plus slides (Fisher Scientific, Pittsburgh, PA, USA) and stored at −20 °C until required.

### 2.4. Tissue Collection for mRNA Expression Analysis in Different Experimental Conditions

Tissue samples were collected at different experimental conditions. Prior to tissue collection, the abalone were anesthetized with 5% MgCl_2_, after which tissue samples were collected. All collected tissues were washed with 1× PBS, immediately flash frozen in liquid nitrogen, and stored at −80 °C until required for total RNA extraction.

#### 2.4.1. Collection of Various Organ Tissues of Pacific Abalone

Ten Pacific abalone of both sexes were sacrificed to collect different organ tissues. The collected tissues were: cerebral ganglion (CG), branchial ganglion (BG), pleuropedal ganglion (PPG), testis (TES), ovary (OVR), digestive gland (DG), heart (HRT), hemocyte (HCY), gill (GIL), mantle (MNT), and muscle (MUS).

#### 2.4.2. Collection of Ganglion Tissues of Pacific Abalone during Gonadal Development

Different ganglion tissues (CG, PPG, BG) of Pacific abalone were collected during different gonadal developmental stages. The gonadal developmental stages of Pacific abalone were determined as previously described [28]. The following developmental stages were evaluated in this study: immature stage (IM), developing stage (DS), ripen stage (RS), and spent stage (SS).

#### 2.4.3. Collection of Ganglion Tissues of Pacific Abalone Conditioned at Effective Accumulative Temperature (EAT)

Two-year-old adult Pacific abalone of both sexes were collected from abalone sea-cage farm from the coast of Jindo-gun, Korea, during early December. All of the collected abalone were at the gonadal recovery phase. After collection, the abalone were transported to the Tou-Jeong Soosan Abalone Hatchery in Dolsan-eup and acclimatized for one week in rearing tanks with continuous aeration and running sea water. To induce early gonadal maturation of abalone, the broodstock is typically conditioned in abalone hatcheries using an effective accumulative temperature (EAT) at 18 °C for approximately 4 months to reach EAT of 1500 °C-days. In the present study, the abalone broodstock were conditioned under EAT exposer as previously described [23]. After acclimatizing at the natural water temperature (7° ± 1 °C), the abalone were kept in an EAT-controlled rearing tank. The water temperature of the tank was steadily increased 1 °C at every alternative day from 7 °C to 18 °C, and finally maintained at 18 °C until the end of the broodstock management period. Ganglion tissues (CG, PPG, BG) were collected at 0 °C-days (i.e., the first day of condition), 500 °C-days, 1000 °C-days, and 1500 °C-days. Five abalones of each sex were sacrificed on each sampling day.

#### 2.4.4. Collection of Ganglion of Pacific Abalone during Induced Spawning Events

Different ganglia (CG, PPG, BG) were collected from abalone at various steps of induced spawning events including heat induction (HI), induction with UV-irradiated water (UV), during spawning (DSW), and post-spawning (PSW). UV- and heat-induced spawning was performed using ripen stage abalone as previously described [23]. Briefly, the abalone at the gonadal ripen stage were first placed under direct sunlight at shell-down orientation for 1 h and shell-up orientation for half an hour. The abalone were then transferred to a spawning tank and induced with UV-irradiated seawater with moderate aeration. Five randomly selected abalone of both sexes were sacrificed at each stage of induced spawning events and ganglion tissues were collected. Finally, after 24 h of spawning, post-spawning samples were collected.

#### 2.4.5. Collection of Different Embryonic and Larval Developmental Stages Samples of Pacific Abalone

After spawning induction, sperm and eggs were collected and mixed carefully in a 30 L bowl with filtered seawater (FSW) to promote successful fertilization. The water temperature of the bowl was maintained at 20 °C. The fertilized eggs were then washed three times with FSW (20 °C), after which the metamorphosis was undergo through different embryonic and larval stages. Samples were collected at the following embryonic and larval developmental stages: fertilized egg (FEG), morula (MOR), blastula (BLU), gastrula (GAS), trochophore larvae (TRL), early veliger larvae (EVL), operculate veliger larvae (OVL), late veliger larvae (LVL), pre-settlement larvae (PSL), and post larvae (PLV).

### 2.5. RNA Extraction and cDNA Synthesis

Total cellular RNAs were extracted from all collected tissues using the ISOSPIN Cell & Tissue RNA kit (Nippon Gene, Tokyo, Japan). First-strand cDNAs were synthesized from 1–4 μL of total RNA depending on the concentration of the total RNA using Superscript III First-strand cDNA synthesis kit (Invitrogen, USA). RACE (5′- and 3′-) cDNAs were synthesized from the total RNA of the CG using the SMARTer^®^ RACE 5′/3′ Kit (Takara Bio Inc., Shiga, Japan). RNA extraction and cDNA synthesis were performed following the protocols provided by the manufacturer of the aforementioned kits.

### 2.6. Cloning and Sequencing of the Full-length FMRF2 Gene (Hdh-FMRF2) in Pacific Abalone

#### 2.6.1. Cloning of Partial Sequence

To obtain a partial fragment of *Hdh-FMRF2* gene, reverse transcription polymerase chain reaction (RT-PCR) was performed using a cDNA template obtained from the CG, forward and reverse primers (Appendix A), and Phusion^®^ High-Fidelity DNA Polymerase (New England Biolabs Inc., Ipswich, MA, USA). The primers were designed from the known FMRF2 cDNA sequence of *Haliotis asinina* (GenBank Accession No. ACD65488). The cloning and sequencing of the partial fragment of the FMRF2 nucleotide was conducted using the TOPcloner™ Blunt kit (Enzynomics, Seoul, Republic of Korea) contains pTOP Blunt V2 vector and DH5α competent cells as previously described [28]. The thermal cycling conditions of the RT-PCR were the following: initial denaturation at 95 °C for 3 min; followed by 36 cycles of denaturation at 95 °C for 30 s, annealing at 58 °C for 1 min, and extension at 72 °C for 45 s; and a final extension step at 72 °C for 7 min. Sequencing was carried out at Macrogen (Seoul, Republic of Korea).

#### 2.6.2. Cloning of 5′- and 3′-RACE Sequence

To obtain the full-length sequence of *Hdh-FMRF2*, rapid amplification of cDNA ends (RACE) PCR was performed in both the 5′- and 3′ directions. Cloning and sequencing of 5′- and 3′-RACE fragments of *Hdh-FMRF2* were performed using the SMARTer^®^ RACE 5′/3′ kit (Takara Bio Inc., Shiga, Japan) as previously described [28]. Gene-specific 5′- and 3′-RACE primers (Appendix A) were designed from cloned *Hdh-FMRF2* partial sequence following the recommendations of the SMARTer^®^ RACE 5′/3′ kit. The thermal cycle conditions were also established according to the recommendations of the manufacturer of the kit. Finally, the sequences of the 5′- and 3′-RACE fragments were combined and the overlaps with the initially cloned partial sequence were trimmed to obtain the full-length sequence of *Hdh-FMRF2*.

### 2.7. In-Silico Analysis of Cloned H. discus hannai FMRF2 (Hdh-FMRF2) Sequence

The nucleotide and amino acid sequences of the cloned *Hdh-FMRF2* gene were analyzed using several online tools and bioinformatic software. The predicted amino acid sequence of *Hdh-FMRF2* cDNA was transcribed from the cloned full-length sequence using the EMBOSS Transeq online tool (http://www.ebi.ac.uk/Tools/st/emboss_transeq/; accessed on 5 August 2022). Open reading frames (ORFs) and potential protein encoding segments were predicted with ORFfinder (https://www.ncbi.nlm.nih.gov/orffinder/; accessed on 5 August 2022). The molecular weight and theoretical isoelectric point (pI) of the protein were computed using the ProtParam (https://web.expasy.org/protparam/; accessed on 5 August 2022) and Protcomp 9.0 (http://www.softberry.com/berry.phtml; accessed on 5 August 2022) online tools, respectively. the functional domains and motifs of the protein were determined using Motif scan (http://myhits.isb-sib.ch/cgi-bin/motif_scan; accessed on 5 August 2022), SMART (http://smart.embl-heidelberg.de/; accessed on 5 August 2022), or InterProScan (http://www.ebi.ac.uk/InterProScan/; accessed on 5 August 2022). The gene ontology (GO) of Hdh-FMRF2 proteins was predicted using the Contact-guided Iterative Threading ASSEmbly Refinement (C-I-TASSER) protein structure prediction server (https://zhanggroup.org/C-I-TASSER/; accessed on 21 July 2022). The protein homology of Hdh-FMRF2 was analyzed using the Basic Local Alignment Search Tool (BLASTP; http://www.ncbi.nlm.nih.gov/BLAST/; accessed on 7 August 2022). Conserved motifs in different Hdh-FMRF2 and related FaRP amino acid sequences were discovered using the Multiple Em for Motif Elicitation (MEME) v. 5.4.1 online tools (http://meme-suite.org/tools/meme; accessed on 7 August 2022).

### 2.8. Analysis of Amino Acid Sequence Alignment and Identity-Similarity Index

Representative amino acid sequences of FMRFamide were obtained from the NCBI protein database (https://www.ncbi.nlm.nih.gov/protein/; accessed on 25 July 2022). Amino acid sequences of FMRFamide from different animals were aligned using ClustalOmega (https://www.ebi.ac.uk/Tools/msa/clustalo/; accessed on 26 July 2022), an online multiple sequence alignment program. The aligned protein sequences were visualized and edited using the Jalview software v. 2.11.1.4. the amino acid sequence identity and similarity of Hdh-FMRF2 with those of other FMRFamide sequence were calculated using the “Idnt & Sim” online suite (https://www.bioinformatics.org/sms2/ident_sim.html; accessed on 10 August 2022). First, the amino acid sequences of the corresponding FMRFamide proteins were obtained from the NCBI protein database, aligned with the Uniprot online alignment tool (https://www.uniprot.org/align/; accessed on 10 August 2022), and finally analyzed with the “Idnt & Sim” suite against the Hdh-FMRF2 amino acid.

### 2.9. Prediction of Three-Dimensional (3D) Structure and Pairwise 3D Structure Alignment

The 3D protein structure of Hdh-FMRF2 and Has-FMRF2 wase generated using an online protein structure prediction program, the Iterative Threading ASSEmbly Refinement (I-TASSER) server (https://zhanglab.ccmb.med.umich.edu/I-TASSER/; accessed on 24 July 2022). The predicted 3D structure and pairwise alignment of the 3D structures of Hdh-FMRF2 and Has-FMRF2 were visualized and analyzed with the UCSF ChimeraX v.1.2.5 software.

### 2.10. Phylogenetic Analysis

A phylogenetic tree was constructed using 54 FaRP protein sequences from different organisms obtained from the NCBI protein database. Phylogenetic analysis was conducted using the alignment build option of the MEGA software (v.11). First, the Hdh-FMFR2 amino acid sequence was aligned with other FaRP protein sequences using ClustalO (https://www.ebi.ac.uk/Tools/msa/clustalo/; accessed on 2 August 2022). Then, a phylogenetic tree was constructed with the MEGA software (v. 11) using the neighbor-joining algorithm with 1000 bootstrap replicates. A detailed list of the FaRP sequences used for phylogenetic analysis is provided in Appendix A.

### 2.11. Peptide Identification Using Nano-LC-ESI-MS/MS

#### 2.11.1. Sample Preparation and Peptide Extraction

Six CG tissues were collected from mature abalone and washed extensively with 1× PBS. Peptide was extracted using EasyPep^TM^ Mini MS Sample Prep Kit (Thermo Fisher Scientific, Rockford, IL, USA) according to kit protocol. Briefly, CG tissues were homogenized with mortar and pestle with lysis solution, and sonicated. Lysate were then centrifuged at 10,000× *g* for 10 min and supernatant were collected. Protein concentration was measured using Pierce^TM^ BCA protein assay kit (Thermo Fisher Scientific, Rockford, IL, USA). Then, 100 µg protein was transferred in a micro tube and volume was adjusted to 100 μL of lysis solution. 50 μL of each reduction solution and alkylation solution were added to protein solution, mixed well and incubated at 95 °C for 10 min. After cooling, protein digestion was performed. To digest the protein, enzyme reconstitute solution and trypsin-protease mix was added to protein solution and incubated at 37 °C for 3 h in a shaking incubator. After incubation, the reaction was stopped by adding digestion stop solution. Later, peptide cleanup was performed using 300 μL digested protein and peptide clean-up column provided in the kit. The eluted peptide was dried by vacuum centrifugation in a speed-vac. The peptide was reconstituted with 100 μL 0.1% formic acid for MS/MS analysis. The peptide concentration was measured using Pierce^TM^ quantitative fluorometric peptide assay kit (Thermo Fisher Scientific, Rockford, IL, USA).

#### 2.11.2. *Nano-LC-ESI-MS/MS* and Peptide Identification

The Nano-LC-ESI-MS/MS analysis was performed using Quadrupole-Orbitrap instrument equipped with Dionex U 3000 RSLCnano HPLC system and Orbitrap Exploris 240 mass spectrometer (Thermo Fisher Scientific, Rockford, IL, USA). Peptide samples were pooled, and peptide fractionation was performed using HPLC system. Peptide fractions were then reconstituted in solvent A (Water/Acetonitrile 98:2 *v*/*v*, 0.1 % Formic acid) and then injected into LC-nano ESI-MS/MS system. Samples were first trapped onan Acclaim PepMap 100 trap column (100 μm × 2 cm, nanoViper C18, 5 μm, 100 Å) and washed with 98% solvent A at a flow rate of 4 μL/min for 6 min, and then separated on a PepMap RSLC C18 column (75 μm × 15 cm, nanoViper C18, 3 μm, 100 Å) at a flow rate of 300 nL/min. The LC gradient was run at 2% to 8% solvent B over 10 min, then from 8% to 30% over 55 min, followed by 90% solvent B for 4 min, and finally 2% solvent B for 20 min. Xcaliber software ver. 4.4 was used to collect MS data. The Orbitrap analyzer scanned the precursor ions with a mass range of 350–1800 m/z with 60,000 resolutions at m/z 200. Mass data are acquired automatically using proteome discoverer 2.5 (Thermo Fisher Scientific, Rockford, IL, USA).

### 2.12. Localization of Hdh-FMRF2 via Fluorescent in Situ Hybridization (FISH)

#### 2.12.1. Synthesis of *Hdh-FMRF2* Riboprobe

Fluorescence antisense and sense riboprobes were synthesized following a previously described protocol [29] with slight modifications. The mRNA probes were prepared from plasmid DNA of 626 bp fragments of the *Hdh-FMRF2* cDNA that was subcloned into the pGEM-T easy vector (Promega, Madison, WI, USA). First, *Hdh-FMRF2* plasmid DNA was linearized using 10 µg of plasmid DNA with the ApaI or SpeI restriction enzymes (Promega, USA) to obtain the antisense and sense probes, respectively. Then, the antisense and sense probes were separately labeled with fluorescein-12-UTP (Roche, Mannheim, Germany) using the T7 or SP6 RNA polymerases (Promega, Madison, WI, USA). The procedures were conducted using 20 μL of reaction mixture containing 1 μg of the linearized plasmid DNA, 4.0 µL of 5× optimized transcription buffer, 2.0 µL of 100 mM DTT, 2.0 µL of fluorescein-12-UTP labeling mix, 2.0 µL of RNase inhibitor, 2.0 µL T7 or SP6 RNA polymerase and RNase-free water (8.0 µL). The reaction mixture was incubated at 37 °C for 2 h. After incubation, the labeled linearized plasmid DNA template was digested at 37 °C for 15 min with DNase I (2.0 µL). The obtained riboprobes were then purified through ethanol precipitation with 1 μL of yeast tRNA (10 mg/mL). The purified riboprobes were stored at −80 °C until required for FISH.

#### 2.12.2. Fluorescence in Situ Hybridization (FISH)

FISH were conducted according to the DIG fluorescein in situ hybridization application manual and a standard protocol previously described [30] with slight modifications. Hybridization buffer was prepared with 25 mL of deionized formamide, 12.5 mL of 20× saline sodium citrate (SSC), 0.5 mL of 10% Tween-20, 0.46 mL of 1M citric acid, and 11.44 mL of DEPC-H_2_O in a total volume of 50 mL. Afterward, hybridization buffer mix was prepared with yeast tRNA at a 9:1 ratio. Cryosections of CG tissue were first air dried for approximately 15 min, then washed with 1× PBS, and prehybridized with hybridization buffer mix at 65 °C for 2 h. After prehybridization, the sections were hybridized overnight with fluorescein RNA probe (300 ng/μL) at 65 °C. The tissue sections were then washed with a hybridization buffer gradient (75%, 50%, 25% volume) mixed with 2× SSC for 10 min each at 65 °C. The sections were then washed with 2× SSC and 0.2× SSC for 15 min each. Next, the sections were washed with a 0.2× SSC gradient (75%, 50%, 25% volume) mixed with PBST (5 min per wash) at room temperature. A final wash was performed with PBST for 5 min at room temperature. After the final wash, the sections were incubated at room temperature with 10% calf serum for 1 h. To detect the hybridization signal, the tissue sections were incubated at room temperature with Anti-Digoxigenin-Fluorescein Fab Fragments antibody (diluted at a 1:500 ratio in 10% calf serum) for 1 h. The tissue sections were then washed with PBST three times at room temperature for 10 min each. Finally, the hybridized sections were counterstained and mounted with VECTASHIELD antifade mounting medium with DAPI (4,6′-diamidino-2-phenylindole) (Vector Laboratories, Inc., Newark, CA, USA). Fluorescent mRNA signals were visualized and captured using a ZEISS LSM 900 confocal microscope with Airyscan2 (ZEISS, Oberkochen, Germany).

### 2.13. Quantitative Real-Time PCR (qRT-PCR) Analysis

The qRT-PCR analysis was performed to quantify the relative mRNA expression of *Hdh-FMRF2* in different experimental tissues. *Hdh-FMRF2* mRNA expression levels were quantified in 11 different tissues (CG, PPG, BG, HRT, OV, TE, DG, HCY, GIL, MNT, and MUS), CG at different gonadal developmental stages (DS, AS, RS, and SS), CG at different EAT °C-days (0, 500, 1000 and 1500 °C-days), CG at induced spawning events, and at different embryonic and larval developmental stages.

The qRT-PCR assay was conducted using the 2× qPCRBIO SyGreen Mix Lo-ROX kit (PCR Biosystems Ltd., London, UK) as previously described [31]. A 20 μL of qRT-PCR reaction mixture was prepared using 1 μL of cDNA template, 1 μL of each gene specific forward and reverse primer, 10 μL of SyGreen Mix, and 7 μL of DDW. Both the target and reference genes were analyzed in triplicate. The PCR conditions were the following: preincubation at 95 °C for 3 min, 40 cycles at 95 °C for 15 s, 60 °C for 20 s, and 72 °C for 20 s. The melting temperature conditions were the following: 95 °C for 10 s, 65 °C for 60 s, and 97 °C for 1 s. At the end of each cycle, a fluorescence reading was recorded for quantification. Florescence amplification and data analysis were performed using a LightCycler^®^ 96 System (Roche, Mannheim, Germany). Relative gene expression was calculated using the 2^−ΔΔCT^ method using the *β-actin* gene of *H. discus hannai* as an internal reference. The expression level in the branchial ganglion was used as a reference value. Specifically, the expression level of the BG was set to 1 and all other data were normalized to this value. All primers used in qRT-PCR analyses are summarized in Appendix A. The qPCR primers of *Hdh-FMRF2* were designed from the unmatched region from nucleotide sequence of *Hdh-FMRF1*, unmatched region was determined by sequence alignment of *Hdh-FMRF2* and *Hdh-FMRF1* using an online multiple sequence alignment program, ClustalW (https://www.genome.jp/tools-bin/clustalw; accessed on 27 July 2022).

### 2.14. Statistical Analysis

The values of mRNA expression of *Hdh-FMRF2* in different organs, CG at different gonadal developmental stages, CG at different EAT °C-days, CG at different steps of spawning events, and different embryonic and larval developmental stages were expressed as mean ± standard error of the mean (SEM). The differences in the relative mRNA expression in different tissues were statistically assessed via nonparametric one-way analysis of variance (ANOVA) followed by Tukey’s post hoc test using the GraphPad Prism 9.3.1 software. Statistical significance was assessed at *p* < 0.05. All graphs were also prepared using the GraphPad Prism 9.3.1 software. Different letters on the bar in the figures indicate significant differences (*p* < 0.05).

## 3. Results

### 3.1. Haliotis discus hannai FMRFamide 2 (Hdh-FMRF2) Sequence

A full-length FMRFamide-related neuropeptide was cloned and sequenced from the CG tissue of Pacific abalone. The cloned FMRFamide-related peptide was termed as *H. discus hannai* FMRFamide *2* (*Hdh-FMRF2*) in this manuscript. The full-length *Hdh-FMRF2* cDNA sequence (GenBank Accession No. MZ224009) was 1256 bp with a poly-A tail. The 5′- and 3′- untranslated regions (UTR) were 133 bp and 508 bp long, respectively (Figure 1). A putative polyadenylation signal (AATAAA) was located at 12 bp upstream of the poly-A tail in its nucleotide sequence. The open reading frame (ORF) of the *Hdh-FMRF2* cDNA sequence was 615 bp, encoding a putative protein of 204 predicted amino acids (NCBI Protein ID: QXP00688).

### 3.2. Features of Hdh-FMRF2 Amino Acid Sequence and Bioinformatic Analysis

The theoretical molecular weight and isoelectric point (pI) of the Hdh-FMRF2 protein were 23.77 kDa and 9.7, respectively. The aliphatic index of the protein was 68.33. The instability index (II) was computed as 47.60, which classified the protein as unstable. The neural nets-nuclear prediction and integral prediction of protein location scores were 1.9 and 9.3, respectively, which predicted the cloned protein as an extracellular (secreted) protein. Predicted GO term analysis using the C-I-TASSER server predicted that the Hdh- FMRF2 protein was significantly linked to the “neuropeptide hormone activity” term (GO:0005184) in the molecular function category with a C-score^GO^ of 0.54 (Appendix A); the “neuropeptide signaling pathway” (GO:0007218) and “regulation of biological process” (GO:0050789) term in the cellular component category with a C-score^GO^ values of 0.571 and 0.82, respectively (Appendix A); and the “cytoplasmic” (GO:0005737) and “extracellular space” (GO: GO:0005615) term in the biological process category with a C-score^GO^ values of 0.51 and 0.69, respectively (Appendix A). The cloned Hdh-FMRF2 protein sequence exhibited the closest similarity to Has-FMRF2 of *H. asinina* (E-value: 2e-122).

The conserved domain search indicated that Hdh-FMRF2 had a signal peptide at amino acid positions 1–23. This protein encoded four copies of FMRFamide-related peptide domain at 51–54 (FLRFa), 62–65 (FLRFa), 89–92 (YLRFa), and 186–189 (FMRFa) amino acids (Figure 1 and Figure 2A). The protein contained a conserved tetrabasic furin-processing site (RRKR) at the 120–123 amino acid position, which separated the amino acid sequence into two domains: the N-terminal region encoding three FaRP tetrapeptides (FLRFamide or YLRFamide) and the C-terminal domain encoding the FMRFamide. Furthermore, the protein had four casein kinase II phosphorylation sites, [S/T]-X(2)-[D/E], located at positions 71–74 (SLLE), 142–145 (SVQE), 149–152 (SSAD), 160–163 (TTDD); an N-myristoylation site, G-{EDRKHPFYW}-X(2)-[S/T/A/G/C/N]-{P}, located at positions 110–115 (GAPNNN); a protein kinase C (PKC) phosphorylation site, [S/T]-X-[R/K], at positions 176–178 (SDR); and an amidation site, X-G-[RK]-[RK], at position 197-200 (FGKR) of the predicted Hdh-FMRF2 protein sequence. Based on all of these features, a Hdh-FMRF2 protein sequence structure is presented in Figure 2A.

Multiple sequence alignment analysis demonstrated that three cysteine residues were conserved at positions 20, 28, and 37 aa of the Hdh-FMRF2 amino acid sequence when aligned with other FMRFamide proteins of *H. asinina* and *H. discus hannai* (Figure 2B). Among the compared invertebrate FMRFamide-related peptide sequences, Hdh-FMRF2 showed the highest identity (85.02) and similarity (89.86) with FMRF2 of *H. asinina* (Table 1). It also showed 28.96% and 37.01% identity and similarity, respectively, to both FMRF1 of *H. discus hannai* and *H. asinina* (Table 1). Identity and similarity index showed that Hdh-FMRF2 and Hdh-FMRF1 have clear differences among their sequences. Table 2 provides a comparative summary of the characteristics of the predicted peptide sequences of Hdh-FMRF2 and Hdh-FMRF1. Peptide prediction clearly showed that there have distinct differences in peptide content among Hdh-FMRF2 and Hdh-FMRF1. A comparative summary of the characteristics of the predicted peptide sequences of Hdh-FMRF2 and Has-FMRF2 is also presented in Appendix A.

The motifs of FMRFamide were parallelly expressed among the compared FMRFamide protein sequences. The motif widths were within 11 and 50 amino acids. Five motifs were identified in Hdh-FMRF2. Likewise, five motifs were identified in the Has-FMRF2 protein, whereas seven motifs were detected in Hdh-FMRF1 and Has-FMRF1 proteins. Five motifs were also recognized in *B. glabrata* among which three motifs matched with Hdh-FMRF2 (Figure 3).

### 3.3. Three-Dimensional Structure and Pairwise 3D Structure Alignment

The 3D structure of the FMRF2 protein in *H. discus hannai* and a pairwise 3D structure alignment of FMRF2 protein of *H. discus hannai* and *H. asinina* are presented in Figure 4A and 4B, respectively. The 3D structures of Hdh-FMRF2 were characterized by several alpha helices separated by loops, forming a helix-loop-helix (HLH) structure. The 3D structure of Hdh-FMRF2 characterized by eight alpha helices was separated by seven loops (Figure 4A) with a C-score of −2.56, an estimated TM-score of 0.42 ± 0.14, and an estimated root-mean-square deviation (RMSD) of 11.3 ± 4.6Å. The pairwise 3D structure alignment of Hdh-FMRF2 and Has-FMRF2 showed a close structural relation (Figure 4B).

### 3.4. Phylogenetic Analysis

To assess the potential evolutionary connections of Hdh-FMRF2 with other FaRPs from vertebrates, annelids, arthropods, and mollusks, a phylogenetic tree was constructed using the neighbor-joining method. The unrooted phylogenetic tree exhibited two major clades, invertebrate FaRP and vertebrate FaRP; invertebrate FaRP further sub-clustered in the invertebrate FLRP and invertebrate FMRF (Figure 5). Hdh-FMRF2 was fitted in the invertebrate FMRFamide clade and sub-grouped with gastropod species. Moreover, Hdh-FMRF2 was fitted with its closest phylogenetic matches, FMRF2 of *H. asinina*.

### 3.5. Identification of FaRP Peptides

CG tissues were examined to detect the presence of mature FaRPs. Representative Nano-LC-ESI-MS/MS clearly showed mass ion peak corresponding to FMRFamide and YLRFamide. Further, NFGEPFLRFa, FGRNFGEPFLRFa, FDSYEDKAYLRFa, SDPGEDP MLKAILLRGAPNNNGWQY and NGWLHFa peptide peaks were also detected (Table 3 and Appendix A).

### 3.6. Fluorescence in Situ Hybridization (FISH) Localization of Hdh-FMRF2

FISH analyses revealed that the *Hdh-FMRF2* riboprobe showed positive signals in neurosecretory cell bodies of the cortex region in the CG when hybridized with the fluorescence antisense probe (Figure 6). In contrast, hybridization with fluorescence sense probe of *Hdh-FMRF2* did not show any signal (Figure 6). These data indicate that *Hdh-FMRF2* mRNA is exclusively expressed in cell bodies in the cortex of the CG in Pacific abalone.

### 3.7. Expression Levels of Hdh-FMRF2 mRNA in Different Tissues of Pacific Abalone

The relative mRNA expression levels of *Hdh-FMRF2* in various tissues of Pacific abalone were analyzed by qRT-PCR and the results are presented in Figure 7. The *Hdh-FMRF2* gene expression was not only exclusively expressed in CG, a lower level of expression was also found in other organs. A negligible expression of *Hdh-FMRF2* was detected in the GIL, DG, MNT, MUS, and HCY. Moreover, the expression levels among male and female were not significantly different.

### 3.8. Expression Levels of Hdh-FMRF2 in CG during Gonadal Development of Pacific Abalone

At different gonadal developmental stages, changes in relative mRNA expression of *Hdh-FMRF2* were observed in the CG of male and female abalone. The expression of *Hdh-FMRF2* was found to be significantly higher in the RS in both sexes (Figure 8). The expression levels of *Hhd-FMRF2* were markedly increased from IM to DS and DS to RS but decreased in SS, and the expression levels among developmental stages were significantly different. A lower mRNA expression of *Hdh-FMRF2* was observed in female at the ripen stage compared to male, however, these changes were statistically insignificant.

### 3.9. Expression Levels of Hdh-FMRF2 in CG during Gonadal Development of Pacific Abalone

The mRNA expression levels of *Hdh-FMRF2* in the CG of EAT conditioned abalone were significantly higher at EAT 1500 °C-days, when the abalone were fully mature (Figure 9). The expression of *Hdh-FMRF2* was significantly increased from EAT 500 °C-days to EAT 1000 °C-days and from EAT 1000 °C-days to EAT 1500 °C-days with the progress of gonadal maturation. However, the differences in mRNA expression among EAT 0 °C-days and EAT 500 °C-days were insignificant when the gonads were in the inactive stage.

### 3.10. Expression Levels of Hdh-FMRF2 mRNA in CG during Induced Spawning Events

During induced spawning events, the mRNA expression levels of *Hdh-FMRF2* were significantly higher at the spawning stage (DSW) and they were significantly downregulated at PSW (Figure 10). Moreover, although a gradual increase in the mRNA expression of *Hdh-FMRF2* was observed among IC, HI and UV stages, the changes were not statistically significant. Interestingly, the mRNA expression levels in the female at the DSW stage were significantly different compared to those of the male. The changes in the mRNA expression levels between the UV stage and DSW stage in the female were not significant, whereas these changes were significant in the male.

### 3.11. Expression Levels of Hdh-FMRF2 mRNA in Embryonic and Larval Developmental Stages of Pacific Abalone

The mRNA expression of *Hdh-FMRF2* was first detected in trochophore larvae (TRL) at 16 h post-fertilization (hpf). Later, it was significantly increased in the EVL stage at 24 hpf. However, the expression levels gradually increased until post-larval stage (PLV) at 144 hpf and significantly higher expression levels were observed at PLV (Figure 11). No mRNA expression was observed in any of the examined embryonic stages.

## 4. Discussion

FMRFamide-related neuropeptides (FaRPs) are important regulators of reproductive regulation, feeding behavior, larval development, and central nervous system development in mollusk species [32]. FaRPs have been reported in all classes of mollusk species, including gastropods, cephalopods, and bivalve mollusks. In the present study, a transcript of FMRFamide gene (*Hdh-FMRF2*) was cloned from Pacific abalone, a gastropod mollusk species. After cloning, it was sought to determine whether this gene is involved in the reproductive regulation and larval development of Pacific abalone.

In this study, a full-length cDNA of *H. discus hannai* FMRFamid 2 (*Hdh-FMRF2*) was sequenced and characterized from the CG of Pacific abalone. Hdh-FMRF2 protein contains four copies of FaRP domain (two copies of FLRFa, one copy of YLRFa and one copy of FMRFa). To designate a peptide as a true FaRP, the following C-terminal sequence is required: an aromatic amino acid, such as phenylalanine (F), tyrosine (Y), or tryptophan (W); an amino acid with a hydrophobic group such as phenylalanine (F), isoleucine (I), leucine (L), methionine (M), threonine (T) or valine (V); and a C-terminal arginine (R) and phenylalanine (F) amide (NH_2_) [18]. The four FaRP domains characterized in the Hdh-FMRF2 met the aforementioned conditions and therefore Hdh-FMRF2 could be classified as a true FaRP. This peptide also contained a highly conserved tetrabasic furin-processing site (RRKR) that separated the peptide into N-terminal and C-terminal region. Previously, an FMRFamide peptide had been reported in Pacific abalone, which was designated as FMRF1 (GenBank Accession no. AVW85483.1). The gene cloned in this study is the second transcript of an FMRFamide peptide isolated from the Pacific abalone, therefore, this peptide is designated as *Hdh-FMRF2*. Two transcripts of the FMRFamide peptide have also been reported in several mollusk species, including tropical abalone [2], marsh snail [33], European cuttlefish [34], and great pond snail [35].

Phylogenetic tree analyses revealed two FaRP clades representing invertebrate and Vertebrate FaRPs. The invertebrate FaRP clade was further sub-clustered into the invertebrate FMRFamide subfamily and the invertebrate LFRFamide subfamily. As expected, the organization of the invertebrate FaRPs in a separate clade in the phylogenetic tree supported their evolutionary relationship. The Hdh-FMRF2 was fitted in the invertebrate FaRP clade and was grouped with almost all mollusk species. It subclustered with its closest relative gastropod mollusk species, *H. asinina* and *Gigantopelta aegis*, revealing that this neuropeptide has a tight evolutionary relationship with gastropod species.

Fluorescence in situ hybridization localization in adult Pacific abalone revealed that the *Hdh-FMRF2* mRNA antisense probe expressed positive signals in the neurosecretory cell bodies in the CG. In situ hybridization and immunohistochemical localization of FMRFamide in adult ganglion tissues and the developing central nervous system have been reported in several mollusk species including tropical abalone [2], common Chinese cuttlefish [13], longfin squid [15], pygmy squid [36] and common octopus [37]. With the analysis of the distribution of RFamide peptide in the central nervous system of mollusks, it has been demonstrated that RFamide peptides localized in the cerebral ganglia are involved in reproduction [14].

The tissue distribution analysis using qRT-PCR revealed that *Hdh-FMRF2* showed significantly higher mRNA expression in the CG of male and female abalone. Although female expressed lower mRNA levels of *Hdh-FMRF2* than the male, these differences were insignificant. Furthermore, a considerable level of expression was observed in the PPG (a ganglion connected to the CG), the heart and the gonads. Negligible levels of mRNA expression were also observed in other tested tissues. In addition to the central nervous system, FMRFamide peptides have been reported in various organs of different mollusk species, such as in the heart of sunray venus clam [5], common Chinese cuttlefish [13], and sea hare [38] as well as in the reproductive organs of common Chinese cuttlefish [13] and garden snail [39].

*Hdh-FMRF2* mRNA expressions were detected in the CG of all stages of gonadal development. The expression was progressively upregulated from the IM to the RS stages, but it was downregulated at the SS. A similar expression pattern has also been reported in the brain of pharaoh cuttlefish [7]. *Hdh-FMRF2* expression was significantly higher at the RS in both sexes. Female abalone exhibited lower expression levels than male; however, these differences were insignificant. The changes in mRNA expression between the DS and RS in female were also not significant. A similar mRNA expression pattern was observed in the CG of EAT-conditioned abalone as observed in the normal gonadal development process. The mRNA expression levels of *Hdh-FMRF2* were increased gradually from EAT 0 °C-days (immature) to EAT 1500 °C-days (fully mature) with the progress of gonadal maturation. The mRNA expression levels reached in a peak at EAT 1500 °C-days. The female exhibited lower expression levels than the male; however, these differences were not significant. The changes in mRNA expression of *Hdh-FMRF2* between EAT 1000 °C-days and EAT 1500 °C-days in female were also not significant. These insignificant changes were also observed during the normal gonadal developmental process of female between DS and RS. The involvement of FMRFamide or FaRP in gonadal development and reproduction has been reported in several invertebrate species. For example, differential expression has been reported in the brain of pharaoh cuttlefish among stage IV and stage V; and female showed significantly lower expression levels at stage V than the male, whereas the opposite trend was observed at stage IV [7]. FMRFamide immunoreactivity was also found in the reproductive ducts of octopus, suggesting that FMRFamide might participate in the central and peripheral peptidergic control of reproduction of common octopus [8]. Moreover, brain FMRFamide influences the secretory activity of the optic glands that control the maturation of the reproductive system in common octopus [40]. Previous studies have also demonstrated that FMRFamide might control ovulation, egg movement and oviposition in the kissing bug [41], as well as play a modulatory role in the locomotion and reproduction of roundworm [42]. Collectively, these findings suggest that *Hdh-FMRF2* is a reproduction-associated peptide in abalone and is involved in the reproductive maturation process.

During induced spawning of Pacific abalone, the mRNA expression of *Hdh-FMRF2* in the CG reached its peak at spawning time or during gamete release (DSW) compared to the other stages. However, female abalone showed significantly lower mRNA expression levels than male at the DSW stage. The changes in mRNA expression levels between the UV and DSW stages in the female were not significant, however, these changes were significant in the male. The mRNA expression levels were significantly downregulated at the PSW stage compared to DSW in both sexes. A previous study reported that expression of FMRFamide significantly decreased just prior to spawning in female tropical abalone, whereas the opposite was observed in the male [43]. The inhibitory effect of FMRFamide on oviposition has been demonstrated in several mollusk species including the great pond snail [44], sea hare [45] and tropical abalone [43]. Moreover, previous studies have confirmed that FMRFamide expression significantly decreased at the time of spawning in female tropical abalone compared to the male [43], which is consistent with the results of the present study. The subsequent downregulation or insignificant increase of FMRFamide during spawning time in females has also been observed in the great pond snail [44] and sea hare [45]. The present findings of lower expression of *Hdh-FMRF2* during gamete release in female abalone compared to male may suggest that *Hdh-FMRF2* might play an inhibitory role in the spawning or release of gametes in female Pacific abalone but not in the male. Similar to the findings of the present study, a previous study also demonstrated that the expression levels of FMRF2 significantly decreased at 24 h of post-spawning in tropical abalone [43]. However, for concrete evaluation, further experiments are needed on peptide quantification and peptide bioassay at spawning stage abalone.

The mRNA expression analysis of *Hdh-FMRF2* during the embryonic and larval development of Pacific abalone revealed that this peptide was not expressed in any of the embryonic stages (fertilized egg, morula, blastula and gastrula). After hatching, the mRNA expression of *Hdh-FMRF2* was detected in trace levels in trochophore larvae. However, *Hdh-FMRF2* expression was significantly increased at the early veliger stage (EVL) when the nervous system began to develop. The expression levels subsequently increased until post-larval stage at which stage the highest expression of *Hdh-FMRF2* was observed. Early studies reported that cerebral ganglia first appeared at the early veliger stage in European abalone [46]. Based on RT-PCR and ISH analyses, it has been reported that *Has-FMRF2* expression was first detected at the trochophore larvae in tropical abalone at 12 hpf and expression was detected in all larval stages until 112 hpf [2]. The present results of the mRNA expression of *Hdh-FMRF2* during the embryonic and larval development stages of Pacific abalone showed similar results as demonstrated in tropical abalone. Furthermore, in situ hybridization and immunohistochemical localization of *FMRFamide* in larval development stages have also been reported in several mollusk species including tropical abalone [2], Pacific oyster [47], nudibranch gastropod sea slug [48], Italian river snail [49] and sea hare [50]. Therefore, the results of the present study suggest that *Hdh-FMRF2* plays a regulatory role in the larval development and nervous system development of Pacific abalone larvae.

## 5. Conclusions

*Hdh-FMRF2*, a second transcript of FMRFamide, was cloned from Pacific abalone. The mRNA expression analyses at different reproductive development experiments revealed that *Hdh-FMRF2* was potentially involved in reproductive maturation. Furthermore, mRNA expression analyses in induced spawning event suggested that *Hdh-FMRF2* inhibits gamete release in female abalone but not in the male. *Hdh-FMRF2* neuronal cell bodies were predominantly localized in the CG. Furthermore, its mRNA expression was observed in all larval developmental stages from the beginning of the nervous system development but not in the preceding embryonic stages, suggesting that *Hdh-FMRF2* regulates the larval development of Pacific abalone larvae. Collectively, the present findings provide important insights into the involvement of specific FaRP (*Hdh-FMRF2*) in reproductive maturation and larval development of Pacific abalone. The results of this study might be valuable for further studies on FaRPs in abalone and useful in broodstock management in abalone hatcheries and aquaculture as well.

## Figures and Tables

**Figure 1 biomolecules-13-00109-f001:**
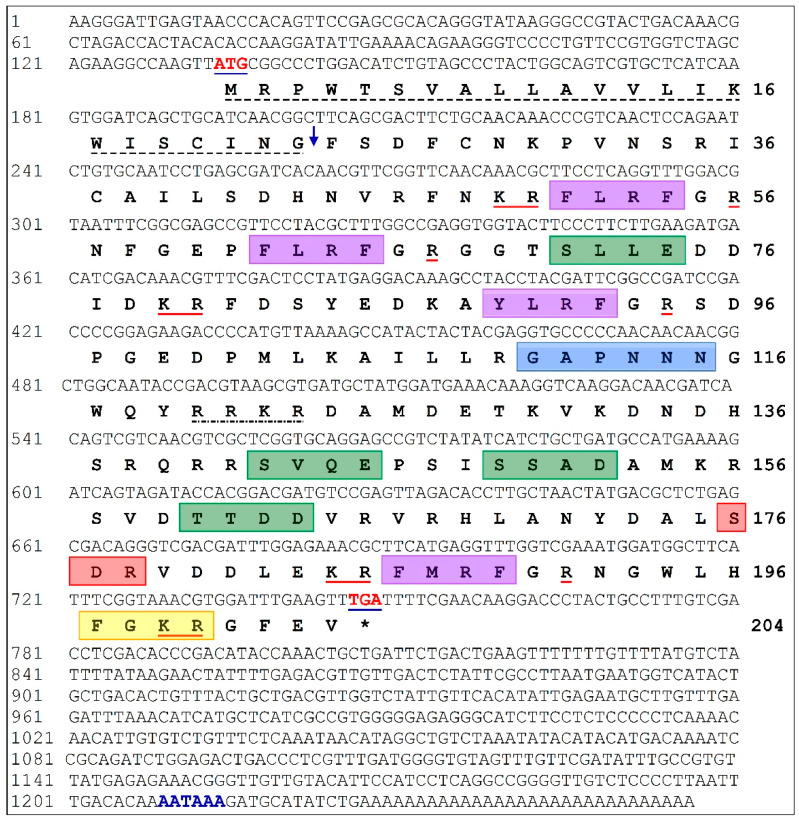
Full length nucleotide and predicted amino acid sequences of Hdh-FMRF2 (GenBank Accession No. MZ224009). The numbers at the right and the left sides indicate the amino acid and nucleotide locations in the sequence, respectively. Start and stop codons (*) are indicated in bold red and blue underlined text. The putative polyadenylation signal (AATAAA) is indicated in bold blue text. The signal peptide is underlined with a stich line in black. The down arrow indicates the predicted signal peptide cleavage site. The predicted FMRFamide related peptide (FaRP) family tetrapeptide motifs are indicated with violet box following the GR processing site. The monobasic and dibasic cleavage sites are underlined in red. A highly conserved tetrabasic furin-processing site (RRKR) is indicated with one dot break line. The green box indicates potential casein kinase II phosphorylation sites. The blue box indicates a N-myristoylation site. The protein kinase C phosphorylation site is indicated with red box. An amidation site is indicated with a yellow box.

**Figure 2 biomolecules-13-00109-f002:**
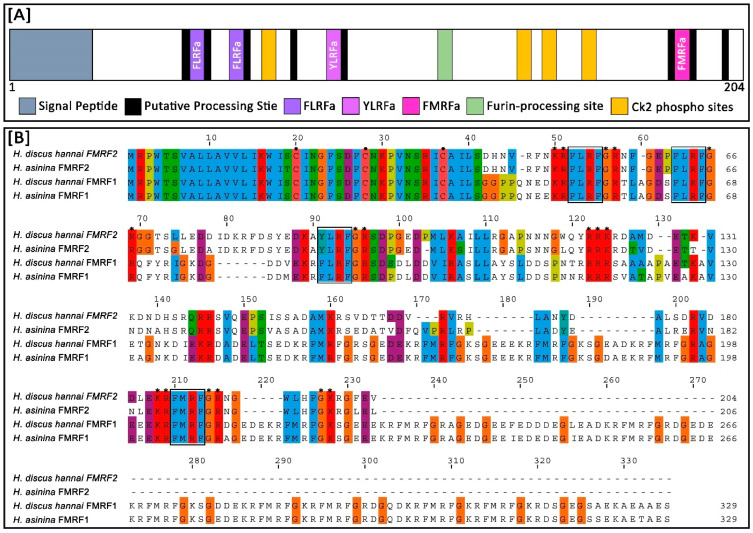
Structure of Hdh-FMRF2 amino acid sequence and multiple sequence alignment of the predicted amino acid sequences of FMRFamide in different abalone species. (**A**) The Hdh-FMRF2 peptide contains two FLRF motifs (violet), one YLRF motif (purple), one FMRF motif (pink), a conserved tetrabasic furin-processing site (mint green), four Ck2 phosphorylation sites (orange), and several putative processing sites flanked by a pair of amino acids (black). (**B**) Multiple sequence alignment of the predicted amino acid sequences of different FMRFamide peptides from *H. discus hannai* and *H. asinine* (details of sequences are presented in Appendix A). The dashes indicate sequence gaps. Three conserved cysteine residues in the aligned sequences are marked with dots. Predicted FaRP domains are marked in black lined box. Processing sites are marked with stars.

**Figure 3 biomolecules-13-00109-f003:**
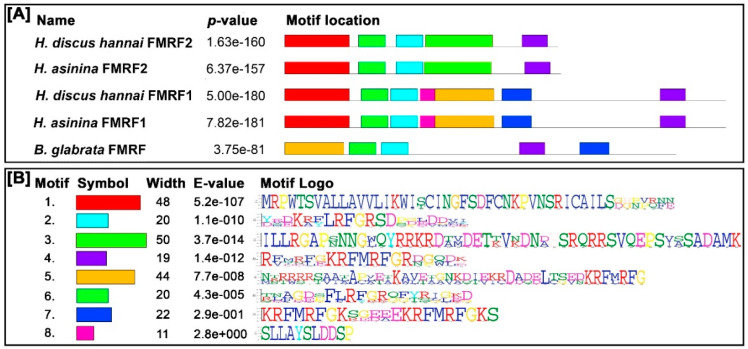
Conserved motif analysis of FMRFamide proteins. (**A**) Schematic diagram of distribution of motifs in FMRFamide peptides (details of sequences are presented in Appendix A). The black solid line represents the length of corresponding FMRFamide protein. Different colored boxes indicate different motifs and their position in each FMRFamide sequence. (**B**) Sequence logo of the FMRFamide motif determined through MEME analysis. The motif logo contains stacks of letters at each position in the motif bits. The height of the individual letters in a stack is the probability of the letter at that position multiplied by the total information content of the stack. The E-value is an estimate of the expected number of motifs that one would find in a similarly sized set of random sequences. The number of widths is the number of amino acids in the respective motif.

**Figure 4 biomolecules-13-00109-f004:**
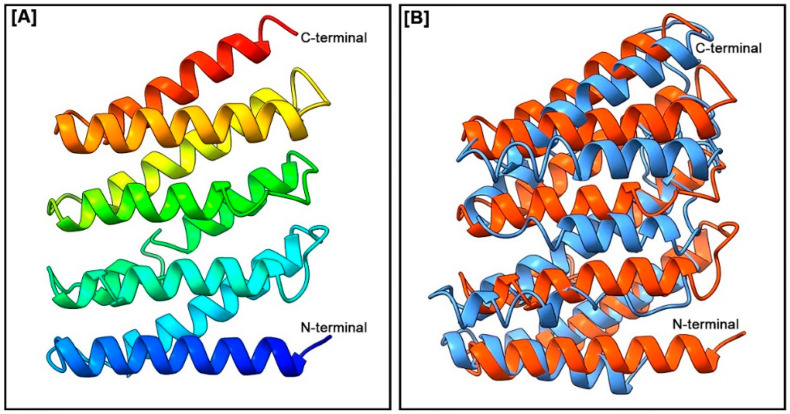
(**A**) Three-dimensional amino acid sequence model of Hdh-FMRF2. The model was constructed using the I-TASSER online tools. Domains between the N-terminal and C-terminal were predicted from the secondary structure. (**B**) Three-dimensional structure alignment of the *H. discus hannai* FMRF2 (in orange) and the *H. asinine* FMRF2 (in sky blue).

**Figure 5 biomolecules-13-00109-f005:**
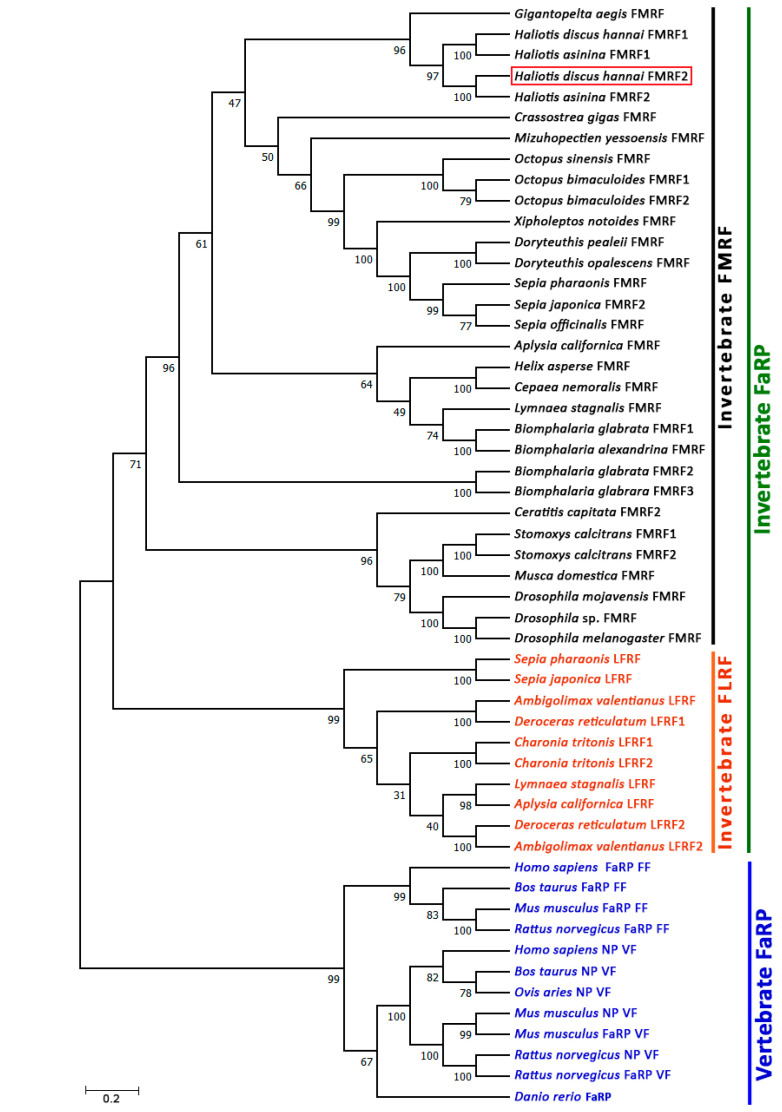
Phylogenetic tree of invertebrate and vertebrate FaRP sequences. Phylogenetic analysis of 53 FaRP amino acid sequences was performed by the bootstrap neighbor-joining method with 1000 bootstrap replicates after aligning the amino acid sequences of FaRPs using the MUSCLE alignment option. The numbers at the nodes indicate bootstrap probability. The scale bar indicates 0.2 units of the expected fraction of amino acid substitutions (1.0 unit = 100 PAMs). The FaRP sequences information used to construct the phylogenetic tree are presented in Appendix A.

**Figure 6 biomolecules-13-00109-f006:**
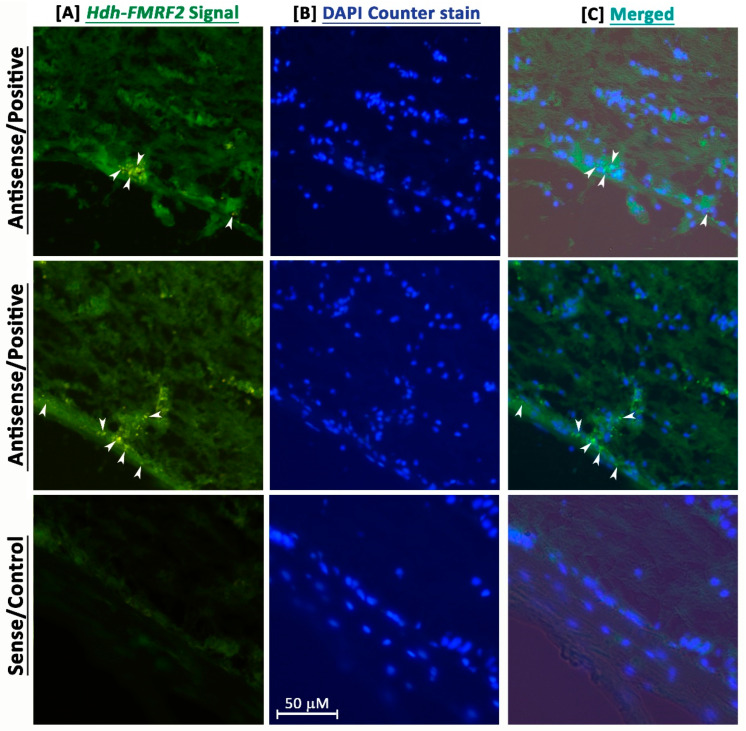
Confocal laser microscopic image of fluorescence in situ hybridization (FISH) of *Hdh-FMRF2* mRNA in the cerebral ganglion of adult Pacific abalone. (**A**) Single confocal optical sections showing positive signal (arrow) of *Hdh-FMRF2* mRNA (green) when hybridized with the anti-sense probe (1st and 2nd row) and no hybridization signal when hybridized with sense probe (3rd row) (**B**) The nuclei are counterstained with DAPI (blue). (**C**) Merged image of (**A**,**B**). Scale bar: 50 μm.

**Figure 7 biomolecules-13-00109-f007:**
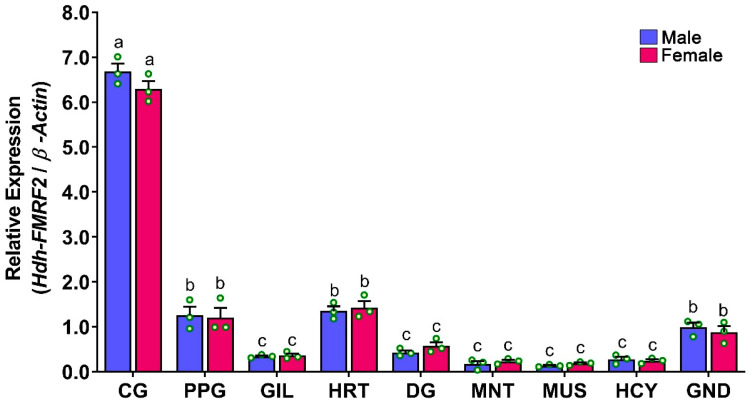
Relative mRNA expression levels (2^–ΔΔCT^) of *Hdh-FMRF2* (mean ± SEM) in different tissues of Pacific abalone in both sexes detected by qRT-PCR. For all bar graphs, the raw data points (green circle) represent biological replicates, the error bars represent the standard error of the mean (SEM), and different letters above the bars indicate significant differences (*p* < 0.05) among organs. CG, cerebral ganglion; PPG, pleuropedal ganglion; GIL, gill; HRT, heart; DG, digestive gland; MNT, mantle; MUS, muscle; HCY, hemocyte; GND, gonad.

**Figure 8 biomolecules-13-00109-f008:**
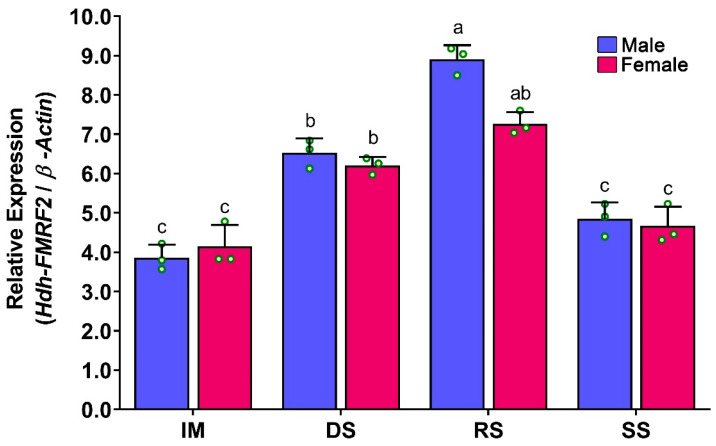
Changes in mRNA expression levels of *Hdh-FMRF2* in cerebral ganglion of different reproductive developmental stages of Pacific abalone in both sexes. For all bar graphs, the raw data points (green circle) represent biological replicates, the error bars represent the standard error of the mean (SEM), and different letters above the bars indicate significant differences (*p* < 0.05) among developmental stages. IM, immature; DS, developing stage; RS, ripen stage; SS, spent stage.

**Figure 9 biomolecules-13-00109-f009:**
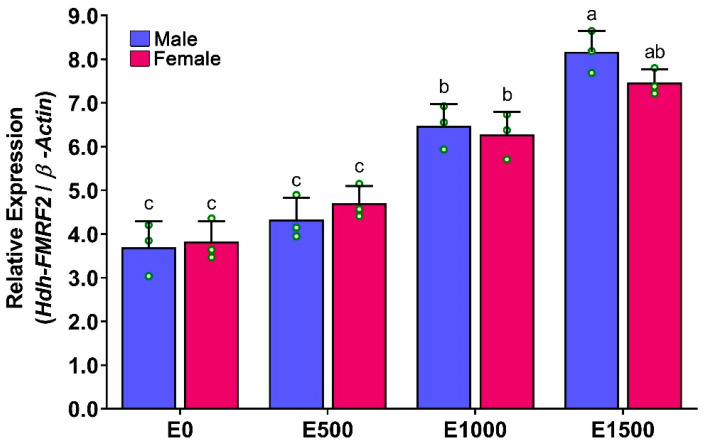
Changes in mRNA expression levels of *Hdh-FMRF2* in the cerebral ganglion of different EAT °C-days of Pacific abalone in both sexes exposed to EAT during broodstock conditioning. For all bar graphs, the raw data points (green circle) represent biological replicates, the error bars represent the standard error of the mean (SEM), and different letters above the bars indicate significant differences (*p* < 0.05) among EAT °C-days. EAT 00, EAT 0 °C-days; EAT 500, EAT 500 °C-days; EAT 1000, EAT 1000 °C-days; EAT 1500, EAT 1500 °C-days.

**Figure 10 biomolecules-13-00109-f010:**
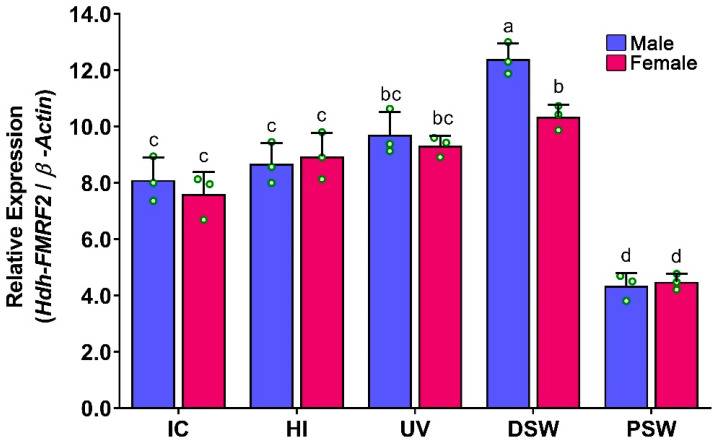
Changes in mRNA expression levels of *Hdh-FMRF2* in the cerebral ganglion during different steps of induced spawning events of Pacific abalone in both sexes. For all bar graphs, the raw data points (green circle) represent biological replicates, the error bars represent the standard error of the mean (SEM), and different letters above the bars indicate significant differences (*p* < 0.05) among the steps of induced spawning event. IC, initial control; HI, heat induced; UV, UV-irradiated water induced; DS, during spawning, AS, after spawning.

**Figure 11 biomolecules-13-00109-f011:**
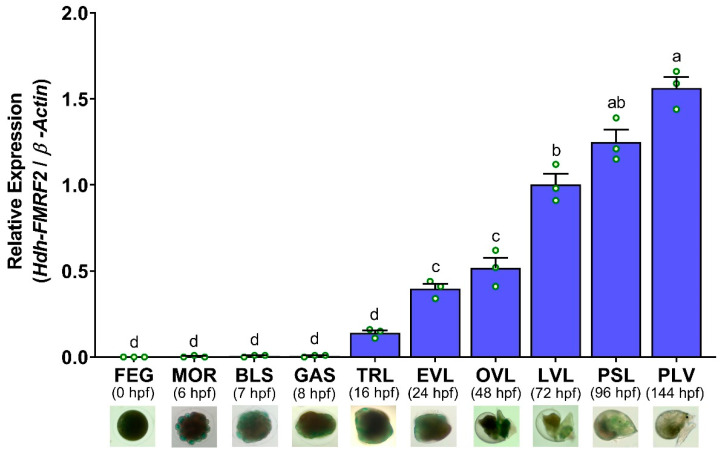
Changes in mRNA expression levels of *Hdh-FMRF2* in different embryonic and larval developmental stages of Pacific abalone. For all bar graphs, the raw data points (green circle) represent biological replicates, the error bars represent the standard error of the mean (SEM), and different letters above the bars indicate significant differences (*p* < 0.05) among different developmental stages. *X*-axis includes names of embryonic and larval development stages, hpf (hours post-fertilization, and microscopic image of each developmental stages. FEG, fertilized egg; MOR, morula; BLU, blastula; GAS, gastrula; TRL, trochophore larvae; EVL, early veliger larvae; OVL, operculate veliger larvae; LVL, late veliger larvae; PSL, pre-settlement larvae; PLV, post larvae.

**Table 1 biomolecules-13-00109-t001:** Amino acid percent identity and percent similarity among FMRF1 and FMRF2 amino acid sequences of *H. discus hannai* and *H. asinina*.

Gene Name	% Identity				% Similarity			
	Hdh-FMRF2	Hdh-FMRF1	Has-FMRF1	Has-FMRF2	Hdh-FMRF2	Hdh-FMRF1	Has-FMRF1	Has-FMRF2
Hdh-FMRF2	100				**100**	**37.01**	37.01	89.86
Hdh-FMRF1	28.96	**100**				**100**	96.96	36.12
Has-FMRF1	28.96	94.53	**100**				**100**	35.82
Has-FMRF2	85.02	27.46	27.46	**100**				**100**

**Table 2 biomolecules-13-00109-t002:** Comparison of peptide predicted between Hdh-FMRF1 and Hdh-FRMF2 precursors of *H. discus hannai*.

Transcript and Peptide Sequences	No. of Amino Acid or Peptides Encoded		Predicted Monoisotopic Mass (kDa)		Isoelectric Point		Attribute
	FMRF1	FMRF2	FMRF1	FMRF2	FMRF1	FMRF2	
Full sequence	329	204	38.38	22.77	9.54	9.70	-
*Pre-tetrabasic cleavage site:*							
FLRFa	2	1	0.58	0.58	–	10.55	Basic
TLAGDSFLRFa	1	×	1.12	–	7.81	–	Neutral
p-QFYRIa	1	×	0.71	–	9.37	–	Basic
Ac-SDSDLDDVIRASLLAYSLDDSPNT	1	×	2.58	–	3.57	–	Acidic
NFGEPFLRFa	×	1	–	1.13	–	6.00	Neutral
FDSYEDKAYLRFa	×	1	–	1.55	–	4.56	Acidic
Ac-SDPGEDPMLKAILLRGAPNNNGWQY	×	1	–	2.76	–	4.56	Acidic
*Post-tetrabasic cleavage site:*							
Ac-SAAAAPAETKAVETGNKDIE	1	×	–	1.97	–	4.41	Acidic
DAMDETKVKDNDHSRQ	×	1	–	1.89	–	4.75	Acidic
FMRFa	13	1	0.60	0.60	10.55	10.55	Basic
NGWLHFa	×	1	–	0.77	–	6.74	Basic

Note: a, amide; p, pyro; ac, acetyl.

**Table 3 biomolecules-13-00109-t003:** Peptides related to Hdh-FMRF2 identified from the cerebral ganglion (CG) of Pacific abalone by LC-MS/MS analysis.

Peptide Sequence	# Proteins	# PSMs	# Missed Cleavages	Theorical MH+ [Da]	XCross
FMRFGR	1	1	1	829.41	0.28
YLRFGR	1	6	1	882.49	0.31
NFGEPFLRFGR	1	2	1	1339.69	0.10
FGRNFGEPFLRFGR	1	4	2	1699.88	0.43
FDSYEDKAYLRFGR	1	3	2	1766.85	0.19
GGTSLLEDDIDKR	1	5	1	14.1871	0.97
SDPGEDPMLKAILLRGAPNNNGWQYR	1	3	2	2912.44	1.02
NGWLHFGKR	1	2	1	1114.59	0.19

## Data Availability

Not applicable.

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
