# Peer review of "Identification and Characterization of Hdh-FMRF2 Gene in Pacific Abalone and Its Possible Role in Reproduction and Larva Development"

_biomolecules, 2023, doi:10.3390/biom13010109_

Round 1
Reviewer 1 Report
The authors cloned the FMRF2 gene from Pacific abalone, Haliotis discus hannai, and investigated the expression patterns in various tissues, developmental stages of gonads, and developing embryos. The experiments were well designed, and the results were clear. However, such kinds of articles had been frequently published in the 1990s and I am concerned about the novelty in some aspects of this manuscript. From a biomolecular perspective, Hdh-FMRFamide has already been identified by transcriptomic and peptidomic analyses in 2019 (Kim et al., 2019, genes), although the entire sequence had not been cloned. From a functional point of view, the biological roles of FMRFamide in reproduction and larval development should not be discussed only by Hdh-FMRF2 mRNA expression without examining the Hdh-FMRF1 expression (Hdh-FMRF1 also encodes FMRFamide peptides). Thus, more critical data in molecular cloning or biological assays are required for publishing in Biomolecules. A few suggestions (major concerns) are shown below.
1. The FMRFamide peptides are also encoded by Hdh-FMRF1 and were detected by mass spectrometry in the previous study (Kim et al., 2019). Identification of Hdh-FMRF2-specific peptides by mass spectrometry will improve the novelty of the manuscript. Additionally, further discussion of similarities and differences between peptides encoded by Hdh-FMRF1 and 2 is required.
2. Related to the point above, the functional roles of FMRFamide peptide should not be evaluated only by Hdh-FMRF2 mRNA expression, given that Hdh-FMRF1 is likely to encode identical peptides to Hdh-FMRF2. The expression levels and patterns of Hdh-FMRF1 mRNA during gonadal and larval development should be compared to that of Hdh-FMRF2. More critically, investigating the effects of peptide treatment for spawning and larval development is of great interest to readers and makes the manuscript more valuable.
Other points are below.
1. “Neuropeptide Hdh-FMRF2” in the title is ambiguous. Which peptide(s) (encoded by Hdh-FMRF2 or FMRF1) potentially regulates reproduction and larval development in the abalone is still unclear.
2. Lines 24-25, the sentence “Collectively, this study provides the first evidence of…” is overstating.
3. Line 106, “various organ tissues” is preferable.
4. Line 340, please confirm the positions (e.g., 142-145, 149-152, etc.).
5. Lines 341-342, I could not understand the G-{EDRKHPFYW}-X(2)-[S/T/A/G/C/N]-{P} for (GAPNNN).
6. Line 354. violet?
7. Line 359, 20, 28, and 37?
8. Line 362, respective to what?
9. Line 366, Has-FMRF2
10. In Fig. 2B, two labels of H.asinina FMRM2 are present.
11. In Table 1, please indicate the means of “Ac-“.
12. Lines 425-426, how can we see the signals are present in neurosecretory cell bodies? Are all of the CG cells neurosecretory? Please explain it in more detail.
13. Line 427, “data not shown” should be avoided according to the "author instructions". Please provide the images.
14. In Fig. 6, please indicate the difference between the top and bottom images.
15. Line 481, PSW?
16. Lines 603-604, the inhibitory roles in the female spawning cannot be discussed only by the Hdh-FMRF2 mRNA expression. The Hdh-FMRF1 expression, bioassay of the peptides, or identification of the peptides is required.
17. Please label A-C in Supplementary Fig. S1.
Author Response
Thank you for reviewing our manuscript and provide valuable suggestions to improve the quality of the manuscript. We have tried to improve the manuscript according to your suggestions. Point by point response to your comments are summarized below and a *doc file also attached herewith.
Major Points:
Point 1: The FMRFamide peptides are also encoded by Hdh-FMRF1 and were detected by mass spectrometry in the previous study (Kim et al., 2019). Identification of Hdh-FMRF2-specific peptides by mass spectrometry will improve the novelty of the manuscript.
Response 1: Thank you for your valuable suggestion. According to your recommendation, we have performed mass spectrometry (LC-MS/MS) analysis and included the results in the revised manuscript (Materials and Methods, Section 2.11 (Pages 5-6, Lines 238-272); Results, Section 3.5, Lines 473-478; Table 3; and Supplementary Figure S2).
Point 2: Additionally, further discussion of similarities and differences between peptides encoded by Hdh-FMRF1 and 2 is required.
Response 2: We have analyzed the identity and similarities of amino acid sequences of Hdh-FMRF2 and Hdh-FMRF1 of H. discus hannai, Ha-FMRF1, and Ha-FMRF2 of H. asinina, and included the data in Table 1 in revised manuscript (Page 10). Further, a comparative analysis of predicted peptides between Hdh-FMRF1 and Hdh-FMRF2 has been presented in Table 2 in revised manuscript (Page 11). Both the amino acid and peptide comparison showed higher dissimilarities among Hdh-FMRF1 and Hdh-FMRF2. As we have included two comparison tables in the revised manuscript, Table 1 (comparison of predicted peptide between Hdh-FMRF2 and Has-FMRF2) of original manuscript has been moved to supplementary material, Supplementary Table S2 in revised manuscript. [Page 10, Lines 404-412]
Point 3: Related to the point above, the functional roles of FMRFamide peptide should not be evaluated only by Hdh-FMRF2 mRNA expression, given that Hdh-FMRF1 is likely to encode identical peptides to Hdh-FMRF2. The expression levels and patterns of Hdh-FMRF1 mRNA during gonadal and larval development should be compared to that of Hdh-FMRF2.
Response 3: Sequence alinement analysis showed that both nucleotide and amino acid sequences of Hdh-FMRF1 and Hdh-FMRF2 have major dissimilarities (alignment figure attached herewith in Page 2: please see attached file). It only showed similarities in the signal peptide region, and other region showed greater dissimilarities. Sequence identity and similarity analysis showed that the amino acid sequence of these two genes have only 28.96% identity and 37.1 similarity (Table 1 in revised manuscript). The qPCR primers were designed from the dissimilar region of nucleotide sequence.
The predicted peptide analysis also showed that these two-sequence (Hdh-FMRF1 and Hdh-FMRF2) contained different peptides. Only one copy of FMRFa exists in Hdh-FMRF2 whereas 13 copies of FMRFa are present in Hdh-FMRF1; one copy of FLRFa is present in Hdh-FMRF2, whereas 2 copies of FLRFa are present in Hdh-FMRF1 (Table 2).
Point 4: More critically, investigating the effects of peptide treatment for spawning and larval development is of great interest to readers and makes the manuscript more valuable.
Response 4: Thank you for your valuable suggestion. We also think that inclusion of this data will improve the quality of the manuscript. However, this experiment should be performed during spawning season which will take longer period to perform. So that, it is not possible to include this data in this manuscript. Please consider this inability.
Minor Points:
Point 5: “Neuropeptide Hdh-FMRF2” in the title is ambiguous. Which peptide(s) (encoded by Hdh-FMRF2 or FMRF1) potentially regulates reproduction and larval development in the abalone is still unclear.
Response 5: With respect to your comment, we have changed the title in the revised manuscript as “Identification and Characterization of Hdh-FMRF2 gene in Pacific Abalone and Its Possible Role in Reproduction and Larva Development”.
Point 6: Lines 24-25, the sentence “Collectively, this study provides the first evidence of…” is overstating.
Response 6: We have changed the sentence by omitting the word “first” in the revised manuscript. [Line 27]
Point 7: Line 106, “various organ tissues” is preferable.
Response 7: As suggested, we have changed this phrase in the revised manuscript. [Line 109]
Point 8: Line 340, please confirm the positions (e.g., 142-145, 149-152, etc.).
Response 8: The position of amino acid checked and corrected as “71–74 (SLLE), 142–145 (SVQE), 149–152 (SSAD), 160–163 (TTDD)” in the revised manuscript . [Page 8; Line 383]
Point 9: Lines 341-342, I could not understand the G-{EDRKHPFYW}-X(2)-[S/T/A/G/C/N]-{P} for (GAPNNN).
Response 8: “G-{EDRKHPFYW}-X(2)-[S/T/A/G/C/N]-{P}” this motif starts with “G”, 2nd amino acid will be anyone without “EDRKHPFYW” (here 2nd bracket means without these amino acids), 3rd and 4th amino acid will be any kind, 5th amino acid will be any one of “S/T/A/G/C/N” (here third bracket means any one from these amino acid), and the 6th amino acid will be any one without “P”. It is available on the website : https://myhits.sib.swiss/cgi-bin/view_mot_entry?name=freq_pat:MYRISTYL
Point 10: Line 354. violet?
Response 10: The typo mistake has been corrected in the revised manuscript. [Page 9, Line 396]
Point 11: Line 359, 20, 28, and 37?
Response 10: The typo mistake has been corrected in the revised manuscript as 20, 28, and 37 aa. [Page 9, Line 402]
Point 12: Line 362, respective to what?
Response 12: This sentence has been rewritten with correct information and revised analysis as “Among the compared invertebrate FMRFamide-related peptide sequences, Hdh-FMRF2 showed the highest identity (85.02) and similarity (89.86) with FMRF2 of H. asinina (Table 1). It also showed 28.96% and 37.01% identity and similarity, respectively to both FMRF1 of H. discus hannai and H. asinina (Table 1).” [Page 9, Lines 404-409]
Point 13: Line 366, Has-FMRF2
Response 13: The typo mistake has been corrected in the revised manuscript. [Page 10, Line 414]
Point 14: In Fig. 2B, two labels of H.asinina FMRM2 are present.
Response 14: The labels in Figure 2B have been changed and the figure also improved with the inclusion of some missing parts.
Point 15: In Table 1, please indicate the means of “Ac-“.
Response 15: In table 1 “Ac-“ means “acetyl”. It has been indicated in the footnote of the table. Table 1 has been moved to supplementary file section as Table S2 in the revised manuscript.
Point 16: Lines 425-426, how can we see the signals are present in neurosecretory cell bodies? Are all of the CG cells neurosecretory? Please explain it in more detail.
Response 16: In abalone, neurosecretory cells are abundantly present in the cortex region of different ganglia [1,2], and neuropeptides [3,4] are secreted from these neurosecretory cells. Several manuscripts illustrated that neuropeptide signals are found in the cortex of ganglia where neurosecretory cells are presented.
- Hahn, K. O. (1994). The neurosecretory staining in the cerebral ganglia of the Japanese abalone (ezoawabi), Haliotis discus hannai, and its relationship to reproduction. Gen. Comp. Endocrinol., 93(3), 295-303.
- Hahn, K. O. (2022). The neurosecretory staining in the pleural-pedal ganglion of the Japanese abalone (Ezoawabi), Haliotis discus hannai, and its relationship to reproduction; with a description of a newly observed neurohemal organ. Compar. Endocrinol., 328, 114106.
- Sharker, M. R., Kim, S. C., Sumi, K. R., Sukhan, Z. P., Sohn, Y. C., Lee, W. K., & Kho, K. H. (2020). Characterization and expression analysis of a GnRH-like peptide in the Pacific abalone, Haliotis discus hannai. Agri Gene, 15, 100099.
- Sharker, M., Kim, S. C., Hossen, S., & Kho, K. H. (2020). Characterization of insulin-like growth factor binding Protein-5 (IGFBP-5) gene and its potential roles in ontogenesis in the Pacific Abalone, Haliotis discus hannai. Biology, 9(8), 216.
Point 17: Line 427, “data not shown” should be avoided according to the "author instructions". Please provide the images.
Response 17: As recommended, negative control data have been added in Figure 6 at 3rd row. [Page 14, Line 488; Page 15, Line 495]
Point 18: In Fig. 6, please indicate the difference between the top and bottom images.
Response 18: As recommended, In Figure 6, top (1st row) and bottom (2nd row) have been labelled in the revised figure.
Point 19: Line 481, PSW?
Response 19: It was typo mistake. “PSW” has be replaced with “ASW”. [Page 17, Line 545]
Point 20: Lines 603-604, the inhibitory roles in the female spawning cannot be discussed only by the Hdh-FMRF2 mRNA expression. The Hdh-FMRF1 expression, bioassay of the peptides, or identification of the peptides is required.
Response 20: According to your suggestion, we have changed the statement as “The present findings of lower expression of Hdh-FMRF2 during gamete release in female compared to male may suggest that Hdh-FMRF2 might play an inhibitory role in the spawning or release of gametes in female Pacific abalone but not in male” and concluded the paragraph with “However, for concrete evaluation, further experiments are needed on peptide quantification and peptide bioassay at spawning stage abalone”.
Point 21: Please label A-C in Supplementary Fig. S1.
Response 21: We have labelled the Supplementary Figure S1 accordingly in the revised manuscript.

Reviewer 2 Report
The manuscript by Sukhan and colleagues cloned and characterized a second transcript of a FMRFamide gene, Hdh-FMRF2, isolated from the cerebral ganglion of Haliotis discus hannai in order to study the expression of this gene and therefore its involvement in reproduction and larval development. This article contributes to your research area by demonstrating a specific temporal involvement of this gene in reproduction and larval development. It is interesting because this gene belongs to a family of neuropeptides described primarily with other functions, such as a cardio excitatory response.
The experiments were well and properly conducted, where first, after cloning and obtaining of the sequence, a series of bioinformatics tools and analyses were performed to categorize this gene as belonging to the FMRFamide-related peptides family. In a second step, ISH experiments confirmed the tissue expression of the mRNA, and finally, through RT-PCR, a modulation of the expression of this gene during reproduction and larval development was demonstrated. The dataset obtained may contribute to the management of this mollusk species.
Specific comments:
1) Show the negative result of the sense probe from the ISH experiment in the figure 6.
2) line 427-429: change the sentence. The Hdh-FMRF2 gene expression was not exclusively in GC, a low level was found in other organs.
3) Check typos: line 39 (familty); line 217 (uniport)
4) Add a comment about the specificity of the primers used for the Hdh-FMRF2 gene in the materials and methods, since this family of genes has very similar products. Would it be possible this primer used to be detecting the FMRF1?
Author Response
Thank you for reviewing our manuscript and provide valuable suggestions to improve the quality of the manuscript. We have tried to improve the manuscript according to your suggestions. Point by point response to your comments are summarized below and a *doc file also attached herewith.
Point 1: Show the negative result of the sense probe from the ISH experiment in the figure 6.
Response 1: As recommended, negative control data have been added in Figure 6 at 3rd row.
Point 2: Line 437-439: change the sentence. The Hdh-FMRF2 gene expression was not exclusively in GC, a low level was found in other organs.
Response 2: According to your recommendation, the sentence has been revised as “The Hdh-FMRF2 gene expression was not only exclusively expressed in CG, a lower level of expression was also found in other organs”. [Page 15, Lines 499-501]
Point 3: Check typos: line 39 (familty); line 217 (uniport)
Response 3: As recommended, typo mistakes have been corrected.
Point 4: Add a comment about the specificity of the primers used for the Hdh-FMRF2 gene in the materials and methods, since this family of genes has very similar products. Would it be possible this primer used to be detecting the FMRF1?
Response 4: According to your suggestion we have included the following sentences in the materials and methods (Section 2.12). “The qPCR primers of Hdh-FMRF2 were designed from the unmatched region from nucleotide sequence of Hdh-FMRF1, unmatched region was determined by sequence alignment of Hdh-FMRF2 and Hdh-FMRF1 using an online multiple sequence alignment program, ClustalW (https://www.genome.jp/tools-bin/clustalw).” [Page 7; Lines 333-337]
Further, Sequence alinement analysis showed that both nucleotide and amino acid sequences of Hdh-FMRF1 and Hdh-FMRF2 have major dissimilarities (alignment figure attached herewith in Page 2; please see the attached file). It showed only 100% similarities in 1-41 aa including signal peptide and other region showed greater dissimilarities. Sequence identity and similarity analysis showed that the amino acid sequence of these two genes have only 28.96% identity and 37.1% similarity (Table 1 in revised manuscript). The qPCR primers were designed from the dissimilar region of nucleotide sequence.
As recommended by another reviewer, we have also included the Identity and Similarity index of FMRF1 and FMRF2 amino acid sequence of H. discus hannai and H. asinina in Table 1.
Furthermore, predicted peptide analysis between Hdh-FMRF1 and Hdh-FMRF2 has also been added in the revised manuscript as Table 2. The predicted peptide analysis also showed that these two sequences contained different peptides. Only one copy of FMRFa exists in Hdh-FMRF2, whereas 13 copies of FMRFa are present in Hdh-FMRF1; one copy of FLRFa is present in Hdh-FMRF2, whereas 2 copies of FLRFa are present in Hdh-FMRF1 (Table 2).

Round 2
Reviewer 1 Report
The authors addressed all of my suggestions appropriately. The manuscript has been well improved and worth publishing in Biomolecules.